# Metabolic Fingerprint of Acromegaly and Its Potential Usefulness in Clinical Practice

**DOI:** 10.3390/jcm8101549

**Published:** 2019-09-26

**Authors:** Betina Biagetti, J.R. Herance, Roser Ferrer, Anna Aulinas, Martina Palomino-Schätzlein, Jordi Mesa, J.P. Castaño, Raul M. Luque, Rafael Simó

**Affiliations:** 1Diabetes and Metabolism Research Unit, Vall d’Hebron Research Institute and CIBERDEM (ISCIII), Universidad Autónoma de Barcelona, 08035 Barcelona, Spain; jorge.mesa@vhir.org; 2Medical Molecular Imaging Research Group, Vall d’Hebron Research Institute, CIBBIM Nanomedicine and CIBERbbn, 08035 Barcelona, Spain; raul.herance@vhir.org; 3Department of Biochemistry, Vall d’Hebron University Hospital, 08035 Barcelona, Spain; roferrer@vhebron.net; 4Endocrinology Department, Hospital Universitari de Vic, 08500 Barcelona, Spain; aaulinasm@gmail.com; 5NMR Facility, Centro de Investigación Príncipe Felipe, 46012 Valencia, Spain; mpalomino@cipf.es; 6Maimonides Institute for Biomedical Research of Córdoba (IMIBIC), 14004 Cordoba, Spainbc2luhur@uco.es (R.M.L.); 7Department of Cell Biology, Physiology, and Immunology, University of Córdoba, 14004 Cordoba, Spain; 8Hospital Universitario Reina Sofía (HURS), 14004 Cordoba, Spain; 9Department of Cell Biology, Physiology, and Immunology, CIBERobn, 14004 Cordoba, Spain

**Keywords:** acromegaly, metabolomics, amino acids, branched chain, insulin resistance, muscular weakness

## Abstract

Insulin-like growth factor-1 (IGF-1) and growth hormone (GH) levels are the main targets for monitoring acromegaly activity, but they are not in close relationship with the clinical course of the disease and the associated comorbidities. The present study was aimed at identifying metabolites that could be used as biomarkers for a better disease phenotyping. For this purpose, metabolic fingerprint using an untargeted metabolomic approach was examined in serum from 30 patients with acromegaly and 30 age-matched controls. Patients with acromegaly presented fewer branched-chain amino acids (BCAAs) compared to the control group (valine: 4.75 ± 0.87 vs. 5.20 ± 1.06 arbitrary units (AUs), *p* < 0.05; isoleucine: 2.54 ± 0.41 vs. 2.80 ± 0.51 AUs; *p* < 0.05). BCAAs were also lower in patients with active disease compared to patients with normal levels of IGF-1 with or without medical treatment. GH, but not IGF-1, serum levels were inversely correlated with both valine and isoleucine. These findings indicate that low levels of BCAAs represent the main metabolic fingerprint of acromegaly and that GH, rather than IGF-1, might be the primary mediator. In addition, our results suggest that the assessment of BCAAs could help to identify active disease and to monitor the response to therapeutic strategies.

## 1. Introduction

Acromegaly (ACRO) is a rare chronic debilitating and multisystem disease due to non-suppressible growth hormone (GH) oversecretion, commonly caused by a pituitary tumor [1]. The autonomous production of GH leads to an increase in the synthesis and secretion of insulin-like growth factor-1 (IGF-1) mainly by the liver, which results in somatic overgrowth, metabolic changes, and several comorbidities [2,3]. Although improvements in surgery and medical therapy decreased the mortality in patients with ACRO, it still remains higher than in the general population [4]. This increase in mortality is mainly due to the higher rate of cardiovascular and malignant diseases in comparison with the age-matched general population [4,5,6,7]. 

The duration of the disease and serum GH/IGF-1 levels were shown as independent predictors of overall mortality and co-morbidities [8,9,10,11,12] and, therefore, current therapeutic strategies are aimed at inhibiting the GH/IGF-1 axis. However, the precise effects that can be attributed to increased GH or IGF-1 levels are still unclear [13,14]. Moreover, both GH and IGF-1 levels are not always directly and unequivocally associated with patients’ quality of life and the presence and severity of co-morbidities [15]. Currently, huge efforts are being made to detect activity disease pointing other variable adjuvants to GH/IGF-1 [16,17]. Overall, these findings suggest that other factors, yet unidentified, could contribute significantly and/or may ultimately be responsible for the different clinical phenotypes, associated comorbidities, and prognosis of these patients. In this context, it is worth noting that one of the major and less understood comorbidities accompanying acromegaly is a profound dysregulation of key players in energy balance and metabolic homeostasis (e.g., glucose and lipid axes).

Metabolomics provides a comprehensive metabolite profile of any biological sample, thus enabling a definition of the chemical phenotype and metabolic fingerprint of human subjects and animal models. This “omics” approach has unique potential for defining metabolites and pathways involved in the pathophysiology of a disease and served to guide the identification of useful biomarkers for monitoring the activity of various diseases, particularly those related to metabolic disturbances. [18]. Hence, it seems conceivable that this approach could be particularly interesting in patients with ACRO in order to complement the information derived from the assessment of GH/IGF-1. 

On this basis, the aims of the present study were to (1) identify metabolites and pathways that could be used as biomarkers of the disease, and (2) explore whether metabolomics could be useful in differentiating those patients with active disease from patients who are either well controlled through pharmacological treatment or already cured. 

## 2. Experimental Section

### 2.1. Ethics Statement

The study was conducted according to the mandates in the Declaration of Helsinki, and the Ethics Committees of the Vall d’Hebron University Hospital approved all procedures (protocol number PR-198/2013). Written informed consent was obtained from all the participants before any action. 

### 2.2. Design and Patients

A case-control study following the Strobe rules was designed [19] A total of 30 patients with ACRO (16 males and 14 females), consecutively recruited from 1 February 2016 to 31 January 2017 at the outpatient clinic of the Endocrinology Department at Vall d’Hebron and 30 age-matched controls were included in the study. The main clinical characteristics of patients with ACRO are displayed in Table 1.

The diagnosis of ACRO and the presence of disease activity was established according to the Endocrine Society Clinical Practice Guideline of Acromegaly [20]. On this basis, we studied three cohorts of ACRO patients; five patients were classified as having active disease (ACRO_active) and 25 presented a controlled disease (14 under medical treatment (ACRO_medical) and 11 without treatment (ACRO_remission)). Baseline characteristics of patients with ACRO are shown in Table 2.

Controlled disease was defined as those patients under medical treatment with IFGF-1 levels within the specific age and sex-adjusted reference range, and those patients who were not under GH receptor antagonist (pegvisomant) in whom random GH concentrations were lower than 1 ng/mL. Cured disease or disease in the remission stage was defined as those patients without medical treatment in whom GH values were ≤0.4 ng/mL after 75 g of oral glucose load. 

At study entry, of those patients with active ACRO disease (*n* = 5), three were under treatment with maximum tolerated doses of first-generation long-acting release (LAR) somatostatin analogues analogues (octreotide-LAR or lanreotide-LAR) and two with pegvisomant, none of them in combination. In the ACRO group controlled by medical treatment (*n* = 14), two were treated with pegvisomant, six with octreotide-LAR or lanreotide-LAR, and two with the combination of pegvisomant + somatostatin analogue, and four were controlled by using cabergoline. All of them had non-curative trans-sphenoidal surgery (TSS) and 10 were also treated with radiotherapy more than 10 years ago (one in ACRO_active, six in ACRO_medical, and three in ACRO_remission group).

The 30 control subjects matched by age, sex, body mass index (BMI), and smoking history were hospital workers, relatives, or friends of the patients with ACRO. Exclusion criteria were pregnancy, mental impairment, type 1 diabetes, active severe diseases, and any other condition or treatment that could influence insulin secretion, provoke insulin resistance, or seriously influence the analysis of the results. Neither patients nor controls were taking protein or BCAAs dietary supplements.

### 2.3. Laboratory Measurements and Phenotypic Characterization

After informed consent was signed, a clinical evaluation, including medical history, smoking status, and physical examination, was performed in all patients and controls. Obesity was defined as BMI ≥30 kg/m^2^. Hypertension was defined as systolic blood pressure >140 mmHg and/or diastolic blood pressure >90 mmHg or the use of antihypertensive medications. Dyslipidemia was defined as total cholesterol (TC) >230 mg/dL, low-density lipoprotein (LDL) >160 mg/dL, triglycerides >150 mg/dL, or treatment with lipid-lowering medications. Homeostatic model assessment of insulin resistance (HOMA-IR) score was calculated as follows: (fasting insulin (mU/L) × fasting glucose (mg/dL)/405) [21], and homeostasis model assessment of β-cell function (HOMA-B) was calculated as follows: (insulin (mU/L) × 360)/(glucose (mg/dL) − 63) [21,22,23].

After overnight fasting of at least 8 h, blood samples were drawn from the cubital vein in the supine position using ethylenediaminetetraacetic acid (EDTA) tubes (K3E) with protease inhibitor (BD Vacutainer–Aprotinin 250KIU Ref. 361017), EDTA tubes (BD Vacutainer K2E (EDTA) 7.2 mg Ref. 368861) and tubes without additive (BD Vacutainer SSTII Ref. 366468).

IGF-I concentrations were measured using a Liaison XL IGF-I chemiluminescence assay (Liaison XL-Diasorin, Italy). The assay is referenced to the first World Health Organization (WHO) International Standard for IGF-I The National Institute for Biological Standards and Control (NIBSC) code 02/254; the limit of detection was 3 ng/mL. GH was measured by ImmuliteXP (Siemens Healthcare, EUA); the assay is referenced to the second WHO NIBSC International Standard 98/574 with analytical sensitivity of 0.01 ng/mL.

### 2.4. Preparation of Plasma Samples for Metabolomics Analysis

A total of 5 mL of peripheral blood freshly extracted was transferred to plasma tubes and allowed to stand for 30 min. The supernatant was then collected and stored at −80 °C until NMR measurement. At the time of analysis, the plasma samples were thawed on ice. Then 300 μL of 10% D_2_O buffer (5 mM propionate-2,2,3,3-d_4_ (TSP), 140 mM Na_2_HPO_4_, 0.04% NaN_3_, pH 7.4) was added to 300 μL of plasma and mixed, and 550 μL was transferred to an NMR tube.

The study followed the recommendations of standards for data analysis in metabolomics [24].

### 2.5. Metabolic Profiling by Nuclear Magnetic Resonance (NMR)

The one-dimensional Carr–Purcell–Meiboom–Gill spectra metabolite resonances (CPMG ^1^H NMR) were acquired using Bruker Avance III 500 MHz spectrometers (Rheinstetten, Germany) running under TopSpin, equipped with a Triple Resonance probe. All runs were carried out using a Bruker Samplecase NMR automation system; prior to each run, the 90° pulse length was determined and set for the run. The field frequency was locked on D_2_O contained in the sample. In all experiments, water suppression was carried out by noise irradiation during the 2-s recycle delay (RD). For all experiments, 64 scans were recorded with an acquisition time of 3 s over a spectral width of 10,000 Hz, and an exponential function was applied to the fragment identificator prior to the Fourier transformation, which resulted in a line broadening of 0.3 Hz. All NMR spectra were automatically phased, baseline-corrected, and referenced to the TSP singlet at 0 ppm, using MestreNova software. Baseline and peak alignment quality control was done by individual verification for each spectrum and, occasionally, a spectrum was manually adjusted.

### 2.6. Statistical Analysis

#### 2.6.1. General Comparisons among Groups

A descriptive analysis was performed to verify the correct introduction of data in the database. Quantitative data are expressed as means and SD (Gaussian distribution) or as medians (p50) and interquartile ranges (IQR; non-Gaussian distribution), and categorical data are expressed as percentages. Data distribution was analyzed by the Kolmogorov–Smirnov test. Logarithmic transformations were performed when necessary to normalize distribution. Comparison between two groups was performed using Student’s *t*-test or Mann–Whitney *U*-test as appropriate. A chi-square test was performed for categorical variables. Fisher’s exact tests were performed when appropriate. Correlations were assessed using Pearson’s correlation coefficient or Spearman rank correlation depending on whether the data were normally distributed. Median values across the three groups (ACRO_active, ACRO_medical, and ACRO_remission) were compared using the Kruskal–Wallis test, and tendencies were compared using the nonparametric Jonckheere–Terpstra test (JT-TT). Statistical analyses were performed using the STATA14 statistical package (USA) for Windows (Serial number: 301406326640). Statistical significance was accepted at *p* < 0.05.

#### 2.6.2. Metabolomic Data

^1^H NMR signals were assigned to metabolites with the help of two-dimensional (2D) experiments, and the Human Metabolome Database (HMBD) 4.0 [25] and Biological Magnetic Resonance Data Bank (BMRB) [26] databases. Concentrations were calculated with the help of the digital electronic reference to access in vivo concentrations (ERETIC) signal that was previously calibrated with a standard sample prepared in identical conditions. Optimal integration regions were defined for each metabolite, and an attempt was made to select the signals without overlapping. Integration was performed with global spectra deconvolution in MestreNova 8.1.

For multivariate analyses, metabolite tables generated from spectra integration were normalized, univariate-scaled, and mean-centered for an easier interpretation of the data and to take the variation of small signals into account. Orthogonal and partial least squares discriminant analysis (OPLS-DA) was performed with SIMCA-P 13.0 (Umetrics, Sweden). Models were validated by permutation and cross-validation analysis. The branched-chain amino acid (BCAAs) concentrations are expressed in arbitrary units (AUs), normalized from mM concentrations.

## 3. Results

### 3.1. Patients

The baseline characteristics of the 30 patients with ACRO and controls are shown in Table 1. Both groups were matched by gender, age, BMI, and smoking habit, and no statistical differences were observed in the presence of other cardiovascular risk factors such as diabetes, hypertension, and dyslipidemia. Likewise, no statistical differences were found in HOMA-IR and HOMA-B, whether when including all subjects or when the analysis was performed by gender.

The distribution of basal characteristics in the group of patients with ACRO according to the disease status is displayed in Table 2. It should be noted that diabetes was present in those patients with active or controlled disease under medical treatment but not in cured patients.

As expected, IGF-1 and GH showed statistically significant differences among groups, being higher in ACRO_active and ACRO_medical compared to ACRO_remission group (*p* < 0.01 in all comparisons). By contrast, we did not find any statistical difference regarding HOMA-IR and HOMA-B among groups. The results of HOMA-IR and HOMA-B remained similar when patients under pegvisomant treatment (*n* = 4) were excluded from the analysis.

### 3.2. Metabolomic Profile

As shown in Figure 1, an OPLS discrimination model separated the patients with ACRO and matched controls on the basis of plasma metabolite differences. The model was validated by permutation and cross-validation.

The main altered metabolites are shown in Table 3. Essential amino acids as a group were lower in patients with ACRO in comparison with control subjects. Among the individual essential amino acids, aside from lower levels of branched-chain amino acids (BCAAs), only lysine was significantly lower in patients with ACRO. Lactate was also lower in the ACRO group, whereas, in contrast, dimethylamine was higher. When these parameters were analyzed based on the status of the disease (Table 4), only lysine and BCAAs (valine and isoleucine) continued being lower in patients with active disease; however, only valine had a positive trend test (JT-TT) (b-Tau 0.30, *p* = 0.04). These results should be interpreted cautiously due to the limited number of patients per group.

Results regarding the Spearman correlation between BCAAs and either GH or IGF-1 are displayed in Figure 2. Serum levels of GH were negatively correlated with both valine (*r* = −0.38, *p* < 0.01) and isoleucine (*r* = −0.56, *p* < 0.001) both in the whole population and in the ACRO group: Valine (*r* = −0.44, *p* <0.02) and isoleucine (*r* = −0.58, *p* < 0.001). We did not find any correlation between IGF-1 or HOMA and BCAAs in either the whole population or in patients with ACRO.

## 4. Discussion

The present comparative metabolomic analysis of patients with or without ACRO suggests that the main metabolic fingerprint of ACRO is a decrease in BCAAs, which seems related to the activity of the disease. This finding might help not only toward a better understanding of the pathophysiology of the disease but in complementing its current therapeutic monitoring.

To the best of our knowledge, this is the first metabolomics study performed in patients with ACRO. However, there is some information supporting our findings. In a metabolomics study performed in non-diabetic patients, Knacke et al. [27] found a negative correlation between IGF-1 and BCAAs. In addition, a clear enhancement of BCAAs was observed in daf-2 mutant worms, which present a clear defect in IGF-1 signaling [28]. In this context, it is important to emphasize that BCAAs are essential amino acids and cannot be synthesized. Therefore, under homeostatic conditions, a precise balance between intake and loss of BCAAs should be maintained. BCAAs play important roles in skeletal muscle growth and homeostasis, are used as an energy source during exercise, and can serve as gluconeogenic precursors [29].

The reason why patients with ACRO present decreased levels of BCAAs remains to be elucidated. GH serum levels were inversely correlated with valine and isoleucine both in the whole population and in patients with acromegaly. In contrast, we did not find any relationship between BCAAs and IGF-1 levels. These findings point to GH rather than IGF-1 as the main mediator of the metabolic fingerprint observed in the present study. In this regard, in a recent study, reduced BCAAs levels were found in two siblings with growth failure due to a pregnancy-associated plasma protein A2 (PAPP-A2) gene mutation, which results in less IGF-1 bioactivity and increased GH secretion [30]. Interestingly, BCAAs levels increased after treatment with recombinant human IGF-1 with concomitant GH reduction.

BCAAs are suggested as biomarkers for insulin resistance and obesity, and even as predictors of diabetes [31,32,33,34,35,36]. Since ACRO is classically associated with insulin resistance, our results would seem to gainsay these previous reports. Of note, however, in our series, we did not observe any difference in HOMA-IR between control subjects and the ACRO group, and, within the ACRO group, HOMA-IR was similar between patients with active and controlled ACRO. In addition, we did not find any relationship between HOMA-IR and circulating BCAAs. Therefore, the metabolic fingerprint observed in patients with ACRO would be unrelated to insulin resistance.

The lack of relationship between ACRO and insulin resistance measured by HOMA observed herein merits specific attention, as it argues against the long-assumed, classical concept that insulin resistance represents the main reason for the high prevalence of type 2 diabetes in patients with ACRO [37,38,39,40,41] In fact, we found less HOMA-B in patients with ACRO than in controls. Medical treatment might influence these results. In this regard, treatment with somatostatin analogues could modify both insulin secretion and insulin sensitivity [42,43,44,45,46], thus making its final result on glucose metabolism unpredictable [42]. By contrast, pegvisomant has favorable effects on glucose metabolism by lowering fasting plasma glucose, HbA1c, and HOMA-IR [47]. However, in a recent, elegant work, in which newly diagnosed and treatment naïve ACRO patients were included, Olarescu et al. found that those patients with high visceral adipose tissue (VAT) presented lower GH levels, higher HOMA-IR, and worse glucose metabolism and lipid profile than patients with normal VAT without insulin resistance [48]. Therefore, it seems plausible that the presence of VAT, rather than GH/IGF-1 activation, could represent the main factor accounting for insulin resistance in patients with ACRO [49]. In addition, a broad range of HOMA-IR values were reported in different series of ACRO but most patients did not present the typical values of HOMA-IR observed in prediabetes or early stages of type 2 diabetes, and a high percentage remained within the normal range [37,48,50,51,52], which argues against the presumed presence of insulin resistance in these patients. Overall, these findings suggest that the underlying molecular mechanisms involved in the increase of diabetes prevalence in ACRO should be carefully revisited.

The reason why the enhancement of GH/IGF-1 induces a lowering of circulating BCAAs seems related to an increased uptake of these amino acids by the muscle, thus favoring an anabolic action. The first studies showing the effect of GH on muscle protein were measured by an isotope dilution technique, which, in fact, assessed the rate of BCAAs disappearance (an index of protein synthesis) [53,54,55,56,57]. These reports, which provided the basis of our knowledge about the role of GH in muscle and confirmed that GH muscle action is mainly anabolic with little effect on proteolysis, consistently showed a BCAAs reduction after GH administration. More recently, Sawa et al. [58] showed that GH administration improves the transport of BCAAs into the myocyte by increasing l-type amino-acid transporter 1 (LAT1) in GH-deficient rats. In line with this, administration of GH to lactating dairy cows increased milk production and, in both plasma and breast tissue, the BCAAs levels were significantly reduced [59]. The authors proposed that the lower concentration of BCAAs was the result of orchestrated events in the lactating gland to optimize the supply of amino acids in relation to the milk formation. Finally, given that GH switches substrate metabolism from glucose and protein utilization to lipid oxidation [60], liver and kidney gluconeogenesis must be enhanced. Gluconeogenic substrates include glycerol, lactate, propionate, and glucogenic amino acids such as valine and isoleucine. Our findings showing lower circulating levels of lactate, valine, and isoleucine in patients with ACRO in comparison with control subjects are consistent with this pathway activation. Again, GH, rather than IGF-1, seems the key player in these metabolic changes. Taken together, this information suggests that both the increased gluconeogenesis and the accelerated consumption of BCAAs can contribute to the low levels of BCAAs detected in acromegalic patients

The decreased serum levels of BCAAs that occur in active ACRO could bear two main consequences, with a paradoxical opposite significance. Firstly, low BCAAs levels could be a contributing factor to the myopathy extensively reported in these patients [61,62]. Secondly, low BCAAs levels could act as protective factor for coronary ischemic disease [63,64]. In this regard, it should be noted that, although cardiovascular disease is the principal cause of premature mortality in patients with ACRO, this is not mainly due to ischemic heart attack [4,5,6,7]. In fact, patients with ACRO present a low incidence of heart ischemic disease despite increased cardiovascular risk factors such as diabetes and hypertension [14,65,66,67,68]. Interestingly, it was recently reported that defective BCAAs catabolism and the chronic accumulation of BCAAs suppress glucose metabolism and exacerbate ischemia–reperfusion injury [64]. By contrast, a role of BCAAs metabolites in cardioprotection against acute ischemia/reperfusion injury was also proposed [69]. Therefore, specific studies addressed at unraveling the cardiovascular consequences of low circulating levels of BCAAs that occur in patients with ACRO are warranted.

In the event that larger studies confirm that the main metabolic signature of ACRO consists of decreased serum levels of BCAAs, a novel avenue in the setting of therapeutic strategies is foreseeable. In this regard, a supplemental intake of BCAAs can be contemplated as a co-adjuvant treatment of patients with ACRO aimed at increasing the muscular strength, including cardiac contractility. In fact, dietary BCAAs supplementation may mitigate muscle soreness following muscle-damaging exercise [70,71] and may also prevent sarcopenia in patients with liver cirrhosis [72,73]. In addition, experimental studies recently showed that BCAAs are essential for maintaining myofibrillar proteins [74], and that regulation of BCAAs catabolism in muscles is important for homeostasis of muscle energy metabolism and, at least in part, for adaptation to exercise training [75]. Nevertheless, further studies aimed at elucidating this important issue in the setting of ACRO are needed.

Our study has several limitations. Firstly, the limited number of patients included warrants that our results should be reproduced in additional larger cohorts. Secondly, the metabolic fingerprint could be masked by the treatment received by patients with ACRO. However, the same metabolic changes were detected when using somatostatin analogues and pegvisomant. This finding, together with the detected inverse correlation between GH levels and both valine and isoleucine, points to the activity of the disease rather than the treatment received as the main factor accounting for the reported metabolic fingerprint. Finally, patients did not follow a standardized diet before samples were obtained for metabolomic analysis, which could increase the variability of the results. Nevertheless, we were able to detect significant differences in BCAAs when comparing ACRO vs. control subjects and active ACRO vs. ACRO controlled under medical treatment.

## 5. Conclusions

In summary, our study suggests that the main metabolic fingerprint of ACRO is a decrease in BCAAs (i.e., valine and isoleucine), which is unrelated to HOMA-IR. These metabolic abnormalities could help to identify active disease and to monitor the response to distinct therapeutic strategies. Moreover, these preliminary findings also uncover a new research area in adjuvant nutritional support in patients with active ACRO. Thus, further studies are warranted to confirm, in a clinical arena, these pilot original results and to explore their mechanistic underpinnings using appropriate experimental models.

## Figures and Tables

**Figure 1 jcm-08-01549-f001:**
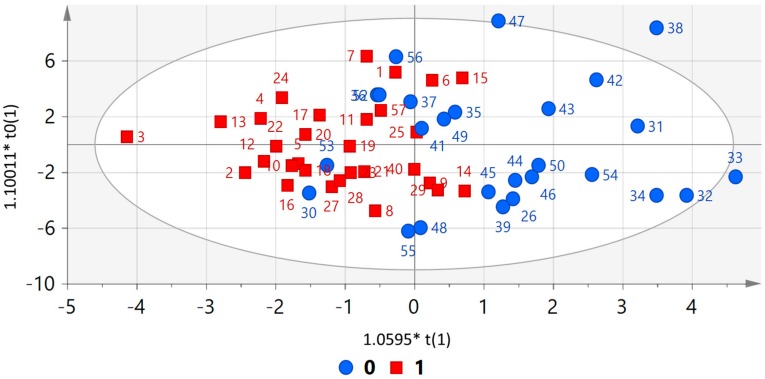
Score plot from orthogonal and partial least squares discriminant analysis (OPLS-DA) of case (1) and control (0) individuals. Input variables are metabolomic data. *R^2^Y* = 0.42, *Q^2^* = 0.16. unit variance scaling. Permutation *R^2^* = 0.217, *Q^2^* = −0.333. cross-validated -ANOVA: 0.066.

**Figure 2 jcm-08-01549-f002:**
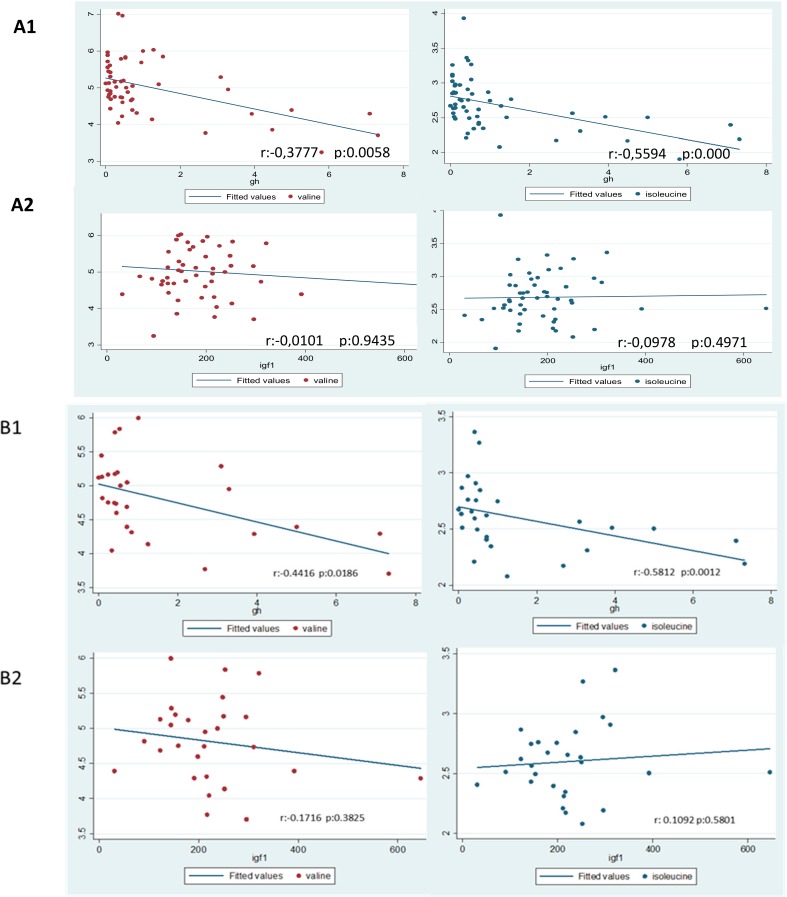
Correlations (**A**) in the whole group and (**B**) in acromegaly (ACRO) patients. *r*: Spearman’s rank correlation coefficient. The figure shows branched-chain amino acid (BCAAs) (valine and isoleucine) Spearman correlations with growth hormone (GH) and insulin-like growth factor-1 (IGF-1) in the whole group (**A1** and **A2**) and in ACRO group (**B1** and **B2**). GH and IGF-1 are expressed in ng/mL and BCAAs are expressed in arbitrary units (AUs).

**Table 1 jcm-08-01549-t001:** Baseline characteristics of acromegaly (ACRO) and matched controls.

	Cases (*n* = 30)	Controls (*n* = 30)	*p*-Value
Gender male (*n*, %)	16 (53.3%)	16 (53.3%)	0.60
Age years (mean +/- SD)	59.5 ± 11.3	56.3 ± 11.	0.22
BMI kg/m^2^ (mean ± SD)	29.57 ± 5.7	27.3 ± 5.1	0.08
Smoking (*n*, %)	5 (16.7%)	5 (16.7%)	0.95
Diabetes (*n*, %)	7 (23.3%)	6 (20.0%)	0.09
Hypertension (*n*, %)	8 (26.7%)	11 (36.7%)	0.58
Dyslipidemia (*n*, %)	11 (36.7%)	14 (46.7%)	0.60
HOMA-IR p50 (IQR)	1.9 (2.8)	3.0 (2.6)	0.06
HOMA-B p50 (IQR)	131.3 (137.1)	164.3 (92.0)	0.25
GH p50 (IQR)	0.63 (2.69)	0.31 (0.71)	0.03
IGF-1 p50 (IQR)	216.1 ± 192	143.0 ± 58	0.00

ACRO: Acromegaly; BMI: Body mass index; SD: Standard deviation; p50: Percentile 50 (median); IQR: Interquartile range; HOMA-IR: Homeostatic model assessment of insulin resistance; HOMA-B: β-cell function; *p*-values: Fisher’s exact test for categorical variables and Mann–Whitney *U*-test for continuous variables.

**Table 2 jcm-08-01549-t002:** Basal characteristic of patients according to disease status.

	ACRO_Active (*n* = 5)	ACRO_Medical (*n* = 14)	ACRO_Remission (*n* = 11)	*p*
Gender Male (*n*, %)	2 (40.0%)	9 (64.0%)	5 (45.0%)	0.55
Age years (mean +/- SD)	55.4 ± 9.2	64.3 ± 11.1	55.4 ± 10.9	0.10
BMI kg/m^2^ (mean ± SD)	29.7 ± 7.4	27.6 ± 4.1	32.1 ± 6.1	0.19
Smoking (*n*, %)	2 (40.0%)	2 (14.3%)	1 (9.1%)	0.28
Diabetes (*n*, %)	2 (40.0%)	5 (37.7%)	0 (0.0%)	0.04
Hypertension (*n*, %)	2 (40.0%)	5 (37.7%)	1 (9.1%)	0.30
Dyslipidemia (*n*, %)	1 (20.0%)	6 (20.0%)	4 (42.9%)	0.88
HOMA-IR p50 (IQR)	2.9 (1.88)	1.8 (3.03)	1.7 (2.57)	0.93
HOMA-B p50 (IQR)	132.1 (114.9)	110.2 (155.8)	135.1 (100.2)	0.60
GH (ng/dL) p50 (IQR)	5.0 (7.9)	0.8 (2.2) ^a^	0.4 (0.5) ^a^	0.00
IGF-1 (ng/dL) p50 (IQR)	392.0 (121)	216.5 (100) ^b^	179.0 (115) ^a^	0.00

ACRO: Acromegaly; BMI: Body mass index; SD: Standard deviation; p50: Percentile 50 (median); IQR: Interquartile range; HOMA-IR: Homeostatic model assessment of insulin resistance; HOMA-B: β-cell function; *p*-values: Fisher’s exact test for categorical and Kruskal–Wallis for continuous variables. ^a^ vs. ACRO_active, *p* < 0.05; ^b^ vs. ACRO_remission, *p* < 0.05.

**Table 3 jcm-08-01549-t003:** Arbitrary units of the main metabolites that differentiate the case and control groups.

	Lysine	Lactate	Valine	Isoleucine	Dimethylamine
ACRO p50 (IQR)	0.72 (0.05)	12.26 (2.78)	4.75 (0.87)	2.54 (0.41)	1.35 (0.18)
Control p50 (IQR)	0.78 (0.08)	14.85 (5.20)	5.20 (1.06)	2.80 (0.51)	1.20 (0.18)
*p*-value Mann–Whitney	0.002	0.001	0.019	0.023	0.016

ACRO: Acromegaly; p50: Percentile 50 (median); IQR: Interquartile range.

**Table 4 jcm-08-01549-t004:** Amino-acid concentration according to disease status.

	1—Active (*n* = 5)	2—ACRO_Medical (*n* = 14)	3—ACRO_Remission (*n* = 11)	Jonckheere–Terpstra	
				Tau_b	*p*
Isoleucine (AUs)	2.50 (0.18)	2.46 (0.44)	2.67 (0.33)	0.26	0.08
Lysine (AUs)	0.69 (0.02)	0.71 (0.08)	0.73 (0.03)	0.28	0.06
Valine (AUs)	4.30 (0.24)	4.71 (1.02)	5.00 (0.54)	0.30	0.04

The table shows median and interquartile range of amino-acid concentrations in each group and the Jonckheere–Terpstra trend test. AUs: Arbitrary units.

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
