# Peer review of "Metabolic Fingerprint of Acromegaly and Its Potential Usefulness in Clinical Practice"

_jcm, 2019, doi:10.3390/jcm8101549_

Round 1

Reviewer 1 Report

The paper titled “Metabolic fingerprint of acromegaly and its potential usefulness in clinical practice” from Biagetti B. and coworkers is well written and of scientific interest, since it shows the results of a metabolomics study in acromegaly patients. The paper has been developed properly, from both methodological and contents point of view. The topic is appealing and original, even if the study included a limited number of patients. Anyway, some comments should be addressed:

Major comments

The study includes a really limited number of patients and among them just five have an active disease; therefore, the author should be more cautious in assessing that the current study provides primary evidence that a decrease of BCAAs represents the main metabolic fingerprint of acromegaly and that this is related to disease activity and possible implications of these findings in the understanding of pathophysiology of acromegaly and in complementing its current treatment should be considered a speculation of the author; The group of acromegaly patients includes subjects under medical therapies which can deeply influence the metabolic status. It is known that somatostatin analogs and pegvisomant have different effect on glucose metabolism, so seems to be interesting a sub-analysis based on acromegaly medical treatment to explore differences in the metabolic fingerprint. This aspect should be better commented by the author.

Minor comments

In the “Experimental Section” there is no mention of adherence to metabolomics reporting data guidelines. (Goodacre R. et al. Proposed minimum reporting standards for data analysis in metabolomics. Metabolomics 2007); In the “Results” the unit of measure of aminoacid concentration is not specified in table 3 and 4; In the “Discussion” the author describes the relationship between acromegaly and insulin resistance and the effects of somatostatin analogs on insulin secretion and sensitivity; it would be appropriate discussing also the effect of pegvisomant on insulin resistance, since it is widely used in the treatment of acromegaly; a recent meta-analysis focused on this specific topic, showing that pegvisomant have favourable effects on glucose metabolism lowering FPG, HbA1c and HOMA-I (Feola T. et al. JCEM 2019). Therefore, the authors should consider the effect of medical treatment both somatostatin analogs and pegvisomant on their findings.

Author Response

The paper titled “Metabolic fingerprint of acromegaly and its potential usefulness in clinical practice” from Biagetti B. and coworkers is well written and of scientific interest, since it shows the results of a metabolomics study in acromegaly patients. The paper has been developed properly, from both methodological and contents point of view. The topic is appealing and original, even if the study included a limited number of patients. Anyway, some comments should be addressed:

Answer: Many thanks for the revision process and your kind comments on our paper.

Major comments

The study includes a really limited number of patients and among them just five have an active disease; therefore, the author should be more cautious in assessing that the current study provides primary evidence that a decrease of BCAAs represents the main metabolic fingerprint of acromegaly and that this is related to disease activity and possible implications of these findings in the understanding of pathophysiology of acromegaly and in complementing its current treatment should be considered a speculation of the author; The group of acromegaly patients includes subjects under medical therapies which can deeply influence the metabolic status. It is known that somatostatin analogs and pegvisomant have different effect on glucose metabolism, so seems to be interesting a sub-analysis based on acromegaly medical treatment to explore differences in the metabolic fingerprint. This aspect should be better commented by the author.

Answers:

 - We have toned down the statement made in the first paragraph of the discussion as recommended. In addition, the different effect of somatostatin and pegvisomant on glucose metabolism has been added to the 5th paragraph of the Discussion.

- The limited number of patients prevented us from performing a robust sub-analysis taking into account the subgroups of medical treatment. Therefore, an eventual influence of the treatment on the metabolic fingerprint cannot be ruled out. This aspect has also been added to the revised manuscript.  Nevertheless, the same metabolic changes were detected when using somatostatin analogues and pegvisomant. This finding together with the detected correlation between GH levels and both valine and isoleucine points to the activity of the disease rather than the treatment received as the main factor accounting for the reported metabolic fingerprint

Minor comments

In the “Experimental Section” there is no mention of adherence to metabolomics reporting data guidelines. (Goodacre R. et al. Proposed minimum reporting standards for data analysis in metabolomics. Metabolomics 2007); In the “Results” the unit of measure of aminoacid concentration is not specified in table 3 and 4; In the “Discussion” the author describes the relationship between acromegaly and insulin resistance and the effects of somatostatin analogs on insulin secretion and sensitivity; it would be appropriate discussing also the effect of pegvisomant on insulin resistance, since it is widely used in the treatment of acromegaly; a recent meta-analysis focused on this specific topic, showing that pegvisomant have favourable effects on glucose metabolism lowering FPG, HbA1c and HOMA-I (Feola T. et al. JCEM 2019). Therefore, the authors should consider the effect of medical treatment both somatostatin analogs and pegvisomant on their findings. 

Answers:

- The adherence to metabolic reporting data guidelines has been added to the revised manuscript as required.

- The aminoacid measurements are represented in arbitrary units (Table 3 and 4) which correspond to mM concentration. This point has been clarified in the revised paper.

- The effect of pegvisomant on IR and the appropriate references have been included in the 5th paragraph of the Discussion. We thank the reviewer this input because it is important to comment on this issue in the setting of this paper.

Reviewer 2 Report

Although vaguely stated I am not completely sure what the advantage is the measurement of BCAAs over looking at serum levels of GH or IGF-1?

In the discussion, the authors made a point of a decrease in BCAA to supply substrates for protein synthesis and for gluconeogenesis.  Do the authors feel that both of these are at work in patients with acromegaly?

More specific comments:

Line 54 - What type of sample?

Line 59 -…particularly interesting in patients with ACRO…

Line 81 – serum IGF-1 levels?

Line 129 - D2O contained

Line 180 – Bold table 2 or not bold table 2?

Figure 2 – units and abbreviation GH or IGF-1

Line 226 – active ACRO?

Line 247 – recombinant human IGF-1

Line 263 – Do we need a year in this reference?

Line 275 - amino acids

Line 279 - amino acids

Line 277 - The rate of BCAA disappearance as a possible indicator of protein synthase or and a substrate for gluconeogenesis are a big part of this study and maybe mentioned in the introduction.  This especially would answer the question why study the levels of BCAA in patients with a metabolic disease such as acromegaly.

Line 299 – patients with acromegaly

Line 300 HTA = hypertension?

Line 315- patients with acromegaly

Author Response

Although vaguely stated I am not completely sure what the advantage is the measurement of BCAAs over looking at serum levels of GH or IGF-1?

Answer: Many thanks for the revision process and your comments on our paper.

- We agree with the reviewer that GH and IGF1 are direct markers of disease activity nevertheless as we comment on the second  paragraph of the Introduction (line 46),  both GH and IGF-1 levels are not always directly and unequivocally associated with patients’ quality of life and the presence and severity of co-morbidities (Webb, 2006). In recent years huge efforts are made to detect activity disease pointing other variables adjuvants to GH/IGF1 ((Giustina et al., 2016; van der Lely et al., 2017). We have added specific comment on this issue to the revised manuscript. In terms of usefulness for clinical monitoring of the disease activity we have toned downs our statements in first paragraph of the Discussion and the Conclusions. However, we feel that our findings not only could be useful in complementing the current therapeutic monitoring but open up a potential new research area in adjuvant nutritional support in patients with active acromegaly.

In the discussion, the authors made a point of a decrease in BCAA to supply substrates for protein synthesis and for gluconeogenesis.  Do the authors feel that both of these are at work in patients with acromegaly?

-The diabetogenic effect of the excess of GH excess is well known. However, the contribution of GH and IGF-1 to hepatic glucose production (HGP) are not completely understood. In the seminal review by Møller and Jørgensen (Endcorcine Reviews 2009) is indicated that enhanced gluconeogenesis (GNG) seems to be promoted in acromegaly. In addition, in vitro experiments have shown increased gluconeogenesis after GH administration in rats (Rogers et al., 1990). Likewise, an interesting study performed by Höybye et al. ( Hormone Metabolic  Research 2008)  in five patients with acromegaly showed that GNG was the main contributor to HGP. In relationship to low levels BCCAs or their accelerated disappearance in acromegaly there is some information in the literature by using isotope dilution (references: from 52 to 56 of the revised paper). Taken together we believe that both the increased GNG and the accelerated consumption of BCAAs can contribute to the low levels of BCCAs detected in acromegalic patients. In order to further clarify this issues this last comment has been added to the revised paper.

The minor specific points have been addressed as required in the revised manuscript.

Reviewer 3 Report

The authors examine the cross-sectional relationship between IGF1 and GH levels and metabolites that could be used as biomarkers.  30 patients with acromegaly and 30 age-matched controls were included. Branched chain amino-acids (BCAAs) were noted to be lower than in controls and in patients with active disease.

The study is of interest.  However, there are some highly significant problems in the design that preclude interpretation as detailed below.

Patients' diet was not controlled.  This is a critical variable, as noted in the limitations by the authors, that highly influences BCAA measurements. Patients were studied in a cross-sectional manner rather than prospective, longitudinal interventional, controlled manner. Most of the differences are of nominal or even lack statistical significance. eg:  table 4. The lack of relationship between GH and IGF-1 and with markers of insulin resistance and BCAA levels raises significance concerns about the reliability of the results shown.  At best, this study is a very preliminary one which should be acknowledged as such with a far more toned down discussion and conclusions.

Author Response

The authors examine the cross-sectional relationship between IGF1 and GH levels and metabolites that could be used as biomarkers.  30 patients with acromegaly and 30 age-matched controls were included. Branched chain amino-acids (BCAAs) were noted to be lower than in controls and in patients with active disease.

The study is of interest.  However, there are some highly significant problems in the design that preclude interpretation as detailed below.

Answer: Many thanks for the revision process and your comments on our paper.

Patients' diet was not controlled.  This is a critical variable, as noted in the limitations by the authors, that highly influences BCAA measurements. Patients were studied in a cross-sectional manner rather than prospective, longitudinal interventional, controlled manner. Most of the differences are of nominal or even lack statistical significance. eg:  table 4. The lack of relationship between GH and IGF-1 and with markers of insulin resistance and BCAA levels raises significance concerns about the reliability of the results shown.  At best, this study is a very preliminary one which should be acknowledged as such with a far more toned down discussion and conclusions. 

Answers:

- The referee is right in indicating that the patients’ diet was not controlled. In fact, as mentioned, we commented on this issue in the last paragraph of the Discussion saying that “Patients did not follow a standardized diet before samples were obtained for metabolomic analysis, which could increase the variability of the results”. Nevertheless, as the reviewer knows, overnight starvation decreases the plasma concentration of most amino acids and normalises any eventual alterations induced by diet (Holecek et al. Nutr Metab 2016; Rousseau et al. Nutrients 2019). In addition, no supplements of BCAAs were received by any of the participants of the study. Consequently, although “not controlled diet” is a theoretical limiting factor, its impact in clinical practice is likely to be negligible. In order to further clarify this issue we have specified the fasting period and the lack of any dietary supplementation in the “Experimental Section” of the revised manuscript.

- We agree with the reviewer that this is not a prospective interventional study but we disagree that the patients were not studied in a controlled manner. In fact, as stated in the “Design and Patients” section this was a case-control study. The recruitment of patients was conducted from February 1st, 2016 to January 31st, 2017 following the Strobe rules required for the case-control studies. We have added these specifications in the revised manuscript in order to avoid any misunderstanding.

- The referee is correct in indicating that the results shown in table 4 do not reach statistical significance in most cases. However, this can be attributed to the fragmentation of the groups according the received treatment. We have added a comment on this limiting factor to the Results section of the revised manuscript.

- The lack of relationship between GH and IGF-1 and HOMA-IR is not surprising, in particular when treated patients are analysed. This important issue has been emphasized in the Discussion of the revised manuscript. As commented, we argued that the metabolic fingerprint observed in acromegalic patients is due to the GH effect rather than changes in insulin resistance. In fact there is abundant evidence that HOMA is not increased in treated  patients with acromegaly (Briet et al., 2019; Ciresi et al., 2018; Olarescu et al., 2016). These studies support the concept that the mechanisms underlying diabetes in acromegaly are not fully understood and are in part different to those involved in classic type 2 diabetes (Ferraù et al., 2018; Olarescu et al., 2016).

- We fully agree that this is a preliminary study and, as recommended, both the discussion and conclusions have been toned down.

Round 2

Reviewer 3 Report

The authors have made a reasonable effort at revising their manuscript.  I have no additional comments at this time.